# How Does FDI Enhance Urban Sustainable Competitiveness in China?

Jian Li [1,2], Lingyan Jiang [1], Bao Jiang [1] and Shuochen Luan [1,*]

1   Department of Economics, Ocean University of China, Qingdao 266100, China; lijian@ouc.edu.cn (J.L.); jianglingyan@stu.ouc.edu.cn (L.J.); jiangbao@ouc.edu.cn (B.J.)
2   Institute of Marine Development, Ocean University of China, Qingdao 266100, China
*   Correspondence: luanshuochen2343@stu.ouc.edu.cn

**Abstract:** Urban sustainable competitiveness (USC) is one of the important indexes to measure the high-quality development of cities in China. Meanwhile, foreign direct investment (FDI) plays the prominent role in improving urban sustainable competitiveness. Therefore, this analysis aimed to test the impact of FDI on the USC and its mechanism using the intermediary effect model with a sample of 282 cities in China during the period 2012–2018. The influencing mechanism includes the scale effect, the technological effect, and the structural effects. The results show that: first, FDI is significantly and positively related to the USC of China, and the scale, technological and structure effects all play a mediating role, with the scale effects being the most significant. Moreover, population size shows negative effect on the USC. Second, the impact of FDI on the USC is regionally heterogeneous. FDI can significantly improve the USC in the eastern region, but has no significant effects on the northeastern, central and western regions. Third, FDI in the eastern region affects USC through structural effects, while the scale and technological effects do not play a mediating role but both effects can directly affect USC. FDI in the northeast region still has the structural effect, but this structural effect does not indirectly affect USC, while FDI in the western region has both scale and structural effects. In addition, the technological and structural effects in the central region have a direct impact on USC, while the scale effect in the western region has a direct impact on USC. Therefore, the findings suggest that utilizing FDI should take into account regional characteristics in China.

**Keywords:** FDI; urban sustainable competitiveness; intermediary effect model

## 1. Introduction

The essence of sustainable development is a stable relationship between human activities and the natural world to ensure that enduring needs are met [1,2]. Urban systems play a vital role in promoting sustainable development as engines of growth, platforms for development and centers of decision making, integrated with ecology, nature and the economy [3]. In recent years, the concept of sustainable urban development has gained popularity among the urban planning community and policy makers [4,5]. In line with this, academic researchers have creatively designed the concept of urban sustainable competitiveness (USC) as a driver of sustainable development [6]. The USC blends the concept of urban sustainability, which is able to measure the sustainability of urban systems, with the concept of urban competitiveness, which reflects a city's ability to allocate resources and services [7]. The introduction of the USC concept further reflects the urgent need to achieve sustainable development. Especially for a country with as many cities as China, it provides new ideas and directions for China to explore and promote high-quality urban development and improve the sustainability level of Chinese cities. According to the Global Urban Competitiveness Report 2019–2020, jointly published by the Chinese Academy of Social Sciences and UN-Habitat, a total of 31 cities in China are ranked among the top

200 in the world [8]. Even so, there is still a large gap between China and European and American cities to a certain extent, and there is still much room for development, especially in terms of quality. Against this backdrop, there is an urgent need for China to tap into its strengths and find practical ways to improve the USC. FDI stands out among the many ways to enhance USC as one of the important ways to promote sustainable development in China [9].

The introduction of FDI can bridge the capital gap, especially from technology-intensive industries, bringing in advanced foreign management techniques and experience, which in turn can improve the USC by increasing the productivity of enterprises [10] and optimizing the industrial structure [11,12]. Moreover, the World Investment Report particularly highlighted FDI as an important source of external finance for achieving the sustainable development goals [13]. The OECD and the United Nations Framework Convention on Climate Change (UNFCCC) consider FDI as a crucial driver of sustainable development, whereas the OECD identifies it as part of the 'Green Economic Opportunities' toolkit, the UNFCCC consider it as a component of the 'Clean Development Mechanism' [14]. According to the statistics of the UNCTAD, China's FDI inflow in 2020 was USD 144.4 billion. In the context of the serious contraction of global trade affected by the pandemic, FDI in China still increased by 4.5%, ranking first in the world [15]. Especially, unique labor advantages and market potential in China can more easily stimulate the inherent potential of FDI to improve the USC. However, FDI also faces the challenge of pollution heaven effect [16,17]. In the case of more relaxed environmental regulations in the host country, the investing country is more willing to transfer local high pollution-intensive industries to the host country. Such industries not only endanger the quality of the local environment, but also do not contribute to the improvement of production efficiency and technological transformation of enterprises, which in turn does not contribute to the sustainable competitiveness of the city. Although FDI may can have a negative impact on environmental quality in some cases, it cannot be denied that FDI is often seen as the remedy to a country's development challenge in most developing countries, as it offers a substitute to domestic investment with the potential to influence the technology endowment and industrial structure of the host country [18]. Therefore, this analysis examines the role of FDI in improving USC and its influence mechanism, so as to explore effective ways to make the efficient use of FDI and avoid the pollution heaven effect, thereby better improving the USC to achieve the Sustainable Development Goals (SDGs).

The rest of our analysis is as follows: Section 2 introduces the related literature review and research hypothesis. Section 3 analyses the underlying methodological framework, data construction. Section 4 reports the empirical results. Section 5 presents conclusions and policy implications.

## 2. Literature Review

### 2.1. The Concept of USC

USC is one of the key components of urban competitiveness, but there is no consensus on the definition of urban competitiveness. Most of the literature defines urban competitiveness from different perspectives, including market share, labor productivity, and urban service functions. Some scholars, such as Lever and Turok [19], considered urban competitiveness as the ability to provide products and services that meet the growing demands of regional, national, business and international markets and achieve sustainable income and stable growth. Porter considered that the output efficiency of local firms is central to regional competitiveness and constructed a diamond model to create an advanced business environment that enhances the output efficiency by developing local firms and attracting efficient firms [20]. In addition, Kresl and Singh gave a series of criteria for evaluating urban competitiveness based on urban service functions, such as high-skilled and high-earning jobs, environmentally friendly products and services, and high employment rates [21]. However, as China's economic development continues to shift from speed growth to qual-

ity and stability, urban competitiveness also needs to consider factors beyond the economy, such as environment, location and culture [22–24].

The emergence of USC, which skillfully combines urban competitiveness and sustainable development, has greatly met the development needs of China's regions. Poot argued that the USC represents a sustained improvement in social welfare, which is related to urban factors that influence sustainable growth [25]. However, it was Balkyte and Tvaronaviciene who first introduced the concept of USC with a broader connotation and combined competitiveness with the dynamics of economic change, social progress and sustainability [6]. Accordingly, the National Institute for Economic Strategy (CASS) has published the USC Index since 2012 and further explains the concept of USC: USC reflects a systematic ability to optimize urban development and meet the complex and critical social well-being of urban residents [7]. At present, research on USC mainly focuses on quantitative analysis, which evaluates the development of a country or region's USC by constructing USC evaluation indicators [26,27]. In short, the USC reflects the ability to grow in the long term and sustainably, rather than short-term economic growth, and focuses on the sustainability of growth patterns to achieve the harmonious development of wealth growth, quality of life and social well-being.

### 2.2. The Relationship between FDI and USC

In a global economy where markets are increasingly interdependent, the role of FDI is crucial as it informs sustainable economic development and innovation policies in many countries and regions [28]. However, due to the relatively new concept of USC, studies on FDI and USC have not directly linked them, but some scholars have examined the role of FDI in promoting sustainable development. For example, Narula first started by constructing a theoretical framework that incorporates the principles of 'Sustainable Investment' (SI) into FDI with the aim of achieving sustainable development goals [29]. Kardos argued that FDI is an important source of sustainable development, while using a combination of analytical methods, data interpretation, and comparisons to verify the importance of green FDI in the sustainable development of EU countries [30]. Aust et al. took the 'Sustainable Development Goals' (SDGs) in Africa as the research object based on the multivariate analysis and an ordered probit model, and found the positive effect of FDI on sustainable development, especially on the local infrastructure and clean energy construction [9]. Furthermore, the adequate disclosure of environmental information plays a crucial role in the positive impact of FDI on sustainable development [31,32]. In addition, several studies highlighted the positive role of FDI in terms of sustainable economies and the environment in particular, including human capital [18], $CO_2$ emissions [33], economic growth [34], energy use [35], FDI connectivity [28], electricity [36], inclusive green growth [14], etc. However, some scholars have also questioned the role of FDI in sustainable development, which focuses on the potential negative impact of FDI on environmental quality [37]. As changes in FDI inflows are highly correlated with the environment [38], FDI from pollution-intensive industries is detrimental to the improvement of environmental quality in host countries, especially for low- and middle-income countries, but the policy environment in the source country can reduce the adverse effects of FDI [39].

Some scholars have also studied the relationship between FDI and competitiveness. However, as the concept of competitiveness was initially applied mainly at the firm level, these studies have also focused on analyzing the role of FDI in the industrial competitiveness. For example, Alvarez and Marin introduced technological innovation and absorptive capacity into the mechanism of the effect of FDI on industrial competitiveness and emphasized that technology creation and absorption are two relevant processes that affect the role that multinational enterprises (MNEs) may play in improving the competitiveness of developing economies [40]. Sekuloska also analyzed technological innovation as an impact mechanism of FDI, but he took national competitiveness as a starting point and found that technological activities play a positive role in FDI to improve national competitiveness [41]. Liu et al. constructed an evaluation system for the green competitiveness of Chinese in-

dustries based on 30 Chinese provinces during the period 2001–2017, and concluded that the quality of FDI had no significant effect on the green competitiveness of industry, but the quantity of FDI had a significant negative effect on the green competitiveness of the industry in neighboring provinces [42].

In conclusion, despite the positive and negative spillover effects of FDI, it does play an important role in sustainable urban development as one of the major sources of external finance. Although there are some conceptual differences between sustainable competitiveness and sustainable development, they both fully embody the concept of sustainable: development that meets the needs of the present without compromising the ability of future generations to meet their needs. Based on this, this analysis considers that FDI also has an effect on USC. Additionally, FDI has distinct regional characteristics due to the different geographical locations, technology levels and market development in the eastern, central and western regions of China. Taking FDI inflows in 2019 as an example, the actual use of foreign investment in the eastern region accounted for 80.79% of the national total, while the central and western regions accounted for 9.32% and 9.89%, respectively. Currently, the eastern region has gained an absolute advantage in FDI inflows by virtue of its geographical location, policies and other advantages. However, due to the absolute scale advantage of the eastern region, a study of the current status of FDI based on these three regions may have obscured the problem of FDI utilization in individual provinces. In other words, the impact of FDI on USC may be regionally heterogeneous. Based on the above judgement, this analysis proposes Hypothesis 1:

**Hypothesis 1.** *FDI significantly affects urban sustainable competitiveness and its effect is regionally heterogeneous.*

*2.3. The Influencing Mechanism of FDI on USC*

Theoretically, the pollution paradise hypothesis confirms the negative spillover effect of FDI; the pollution halo hypothesis confirms the positive spillover effect of FDI; and the environmental Kuznets hypothesis further confirms both effects of FDI from a non-linear perspective [17,43–45]. Despite the differences in the effects of FDI, all these hypotheses examine the impact mechanisms of FDI in terms of scale, technological and structural effects, and these analyses are also applicable to the analysis of the impact mechanisms of FDI on the USC (see Figure 1).

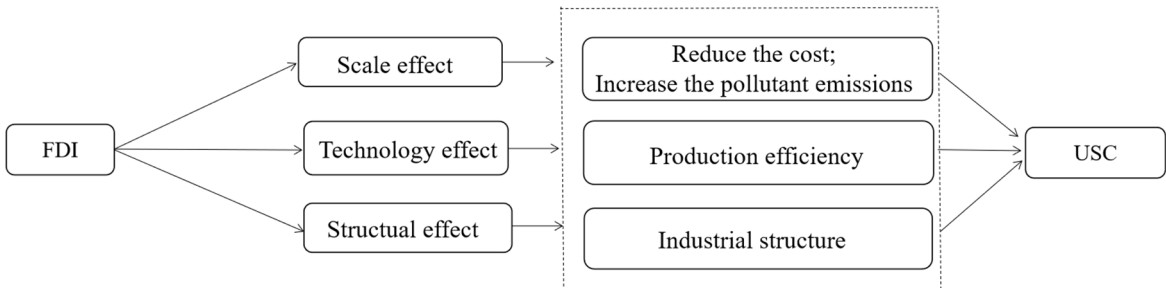

**Figure 1.** Impact of FDI on the USC.

In terms of the scale effect, when a large amount of FDI enters, local cities can take advantage of external development factors such as a good business environment and infrastructure to reduce the production costs of enterprises, enhance their productivity and invest more in the improvement of management methods [46]. Due to the competitiveness being closely related to productivity [47], the cost reductions and efficiency gains caused by scale effects can further improve the USC. But when FDI comes from pollution-intensive industries such as electricity, coal and others, as the size of FDI increases, emissions of sulfur dioxide and carbon dioxide increase accordingly, and firms have to add more environmental input costs to improve the environmental quality in order to reduce this

negative externality [43,44]. However, considering the lagging effect of environmental regulation, it is difficult to improve environmental quality in the short term, which in turn is detrimental to the USC.

In terms of the technological effect, it is related to the technological absorption capacity of cities [48]. For some cities with mature technological capabilities, it is easier for these cities to digest and absorb advanced technologies, improve resource use efficiency and reduce pollutant emissions under the influence of FDI [44]. However, for some cities with less developed technological capabilities, the USC is likely to decrease following large FDI inflows. For example, investing countries protect core technologies or even implement technological monopolies [44]. The technological demonstration effect of FDI is limited, and it is difficult for local firms to learn, imitate and absorb advanced technologies. Moreover, if local firms do not develop strong competitiveness, the entry of FDI can squeeze the market share of domestic firms, and this negative competition can reduce business profits, employment and tax revenue, etc., which in turn affect the USC.

In terms of structural effects, FDI from high-tech intensive industries can direct capital flows to corresponding local industries and integrate various resources of foreign capital and labor to achieve complementary resource advantages of local industries [49]. However, FDI from low-technology-intensive industries can exacerbate the imbalance in the industrial structure. According to the marginal industry expansion hypothesis [43], host countries with weaker economies are more likely to attract foreign FDI in some marginal industries that are or will be disadvantaged, which can undoubtedly pose a threat to local industries, the environment and economic efficiency. In particular, some local industries usually lower their barriers to entry and introduce double-standard FDI to attract foreign firms, a practice that is completely contrary to the concept of sustainable development and detrimental to the USC [17,50]. Accordingly, this analysis proposes Hypothesis 2:

**Hypothesis 2.** *FDI can affect USC through scale effect, technological effect and structural effect.*

The contribution of this article is as follows: on the topic of how FDI affects USC, most studies focus on the impact of FDI on the environmental or economic sustainability [14,17,18,28,33–36], and fewer studies have focused on the impact of FDI on sustainable competitiveness. As the sustainable competitiveness is a comprehensive concept, neither environmental nor economic sustainability can fully capture the sustainable competitiveness. This analysis selects the urban sustainable competitiveness index, which uses a non-linear weighted composite method to measure the USC of China, and carries out empirical analysis in six aspects: knowledge, harmony, ecology, culture, territory-wide, and information, which is authoritative and scientific [7]. Moreover, even if there are studies that integrate various dimensions to construct the indicators of USC, these mostly focus on the qualitative analysis of USC [26,27], ignoring the key role of FDI in it, not to mention the analysis of the inner mechanism of FDI affecting USC. This study uses a mediating effects model to analyze the influencing mechanism of FDI on USC, which can provide new ideas and directions for designing strategies to improve the USC. In view of this, this analysis expands the research scope of the existing literature.

## 3. Research Design

This analysis selects 282 cities in China as the research objective during the period 2012–2018. The empirical study includes three main steps. The first step is to test the impact of FDI on USC. The second step is to test whether FDI has an impact on the USC through the scale effect, technological effect and structural effect, that is, whether the three effects play a mediating role in the effect of FDI on USC. The third step is to perform further analysis, including endogenous analysis and regional heterogeneity.

*3.1. Measurement Model Construction*

This analysis examines the impact of FDI on USC using a mediating effects model. The mediating effect refers to the indirect effect of the explanatory variables on the explanatory variables under the influence of the mediating variables, allowing for the exploration of the underlying mechanisms between them and the integration of existing research or theory to make it more systematic and comprehensive. The advantage of the mediating effect model over other models is that it is able to consider both the direct and indirect effects of the independent variable on the dependent variable, thus providing a more accurate assessment of the total effect of the independent variable on the dependent variable. In addition, the model can be used to explore the effects of multiple mediating variables on the dependent variable, thus providing a more comprehensive understanding of the relationship between the independent and dependent variables. Specific methods refer to the study by Kong et al. [51] and added the explained variable (*usc*), the core explanatory variables FDI (*fdi*), the mediating variables including the scale effect (*scale*), the technological effect (*tech*), and the structural effect (*stru*), and other control variables to the model. In addition, we logarithmized the explanatory, mediating and control variables. Logarithmization reduces the absolute value of the data, facilitates comparisons, calculations, and the nature of the variables and relationships remain unchanged after logarithmic treatment. However, we did not logarithmize the explanatory variables because of the small range of fluctuations in the sample data in the place of the explanatory variables. The intermediate model was constructed as follows:

$$\begin{cases} usc_{it} = a_0 + a_1 lnfdi_{it} + \sum_{k=1}^{6} a_{k+1} lnX_{kit} + \mu_i + \lambda_t + \varepsilon_{it}, \ k = 1, 2 \cdots 6 \\ W_{lit} = b_0 + b_1 lnfdi_{it} + \sum_{k=1}^{6} b_{k+1} lnX_{kit} + \mu_i + \lambda_t + \varepsilon_{it}, \ k = 1, 2 \cdots 6, \ l = 1, 2, 3 \\ usc_{it} = c_0 + c_1 lnfdi_{it} + c_2 lnW_{lit} + \sum_{k=1}^{6} c_{k+2} lnX_{kit} + \mu_i + \lambda_t + \varepsilon_{it}, \ k = 1, 2 \cdots 6, \ l = 1, 2, 3 \end{cases} \quad (1)$$

In Equation (1), *i* and *t* represent listed cities and year, respectively. $usc_{it}$ is the explained variable and represents the urban sustainable competitiveness. $fdi_{it}$ is the core explanatory variable and represents FDI inflows. $W_{lit}$ is the mediating variable and *l* represents the number of mediating variables, including the scale effect $scale_{it}$, the technological effect $tech_{it}$, and the structural effect $stru_{it}$. $X_{kit}$ is a set of control variables and *k* is the number of mediating variables. $\mu_i$ and $\lambda_t$ are the city and year effects, and $\varepsilon_{it}$ is the random disturbance term. In general, three steps are required to detect mediating effects (see Figure 2). Firstly, test whether the coefficient $a_1$ in the first equation is significant. If the coefficient $a_1$ is significant, it is necessary to continue to test the second equation. If the coefficient $a_1$ is not significant, there is no mediating effect. Secondly, test whether the coefficient $b_1$ in the second equation and the coefficient $c_2$ in the third equation are significant. If the coefficients $b_1$ and $c_2$ are significant, this indicates the existence of a partial mediating effect. If at least one of $b_1$ and $c_2$ is not significant, it needs to be further analyzed by Sobel's test. If the coefficient of Sobel's test is significant, there's a mediating effect; otherwise, the mediating effect does not exist. Thirdly, test whether the coefficient $c_1$ in the third equation is significant. If the coefficient $c_1$ is not significant, it indicates the existence of a fully mediating effect.

*3.2. Variables*

3.2.1. Dependent Variable

The urban sustainable competitiveness index was chosen to represent the USC (*usc*) provided by the Annual Report on China's Urban Competitiveness [7]. The urban sustainable competitiveness index is a comprehensive process that includes 54 secondary indicators such as knowledge-based cities, harmonious cities, ecological cities, cultural cities, holistic cities and information-based cities. All data were first standardized according to the equal-weighted summation method, combining the secondary and primary indicators, and then a non-linear weighted composite method was used to obtain an overall score.

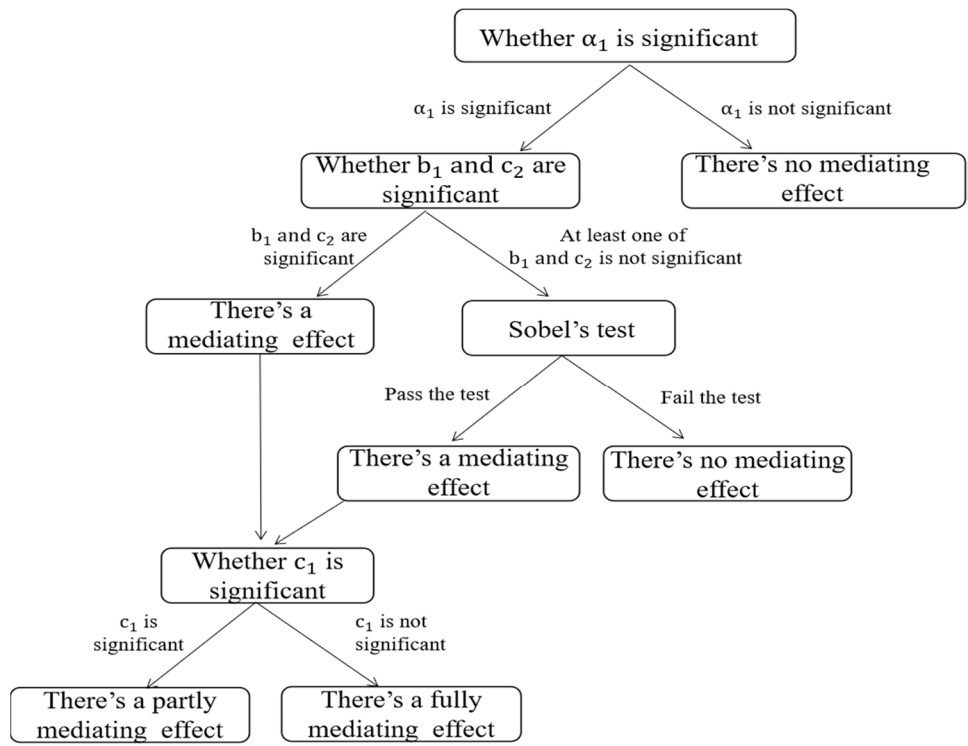

**Figure 2.** Decision tree of the mediating effect.

### 3.2.2. Independent and Intermediary Variables

The variable of FDI ($fdi$) was measured by the amount of foreign capital actually utilized and selected from the China City Statistical Yearbook. Intermediary variables include the scale effect (*scale*), the technological effect (*tech*) and the structure effect (*stru*). The index of industrial sulfur dioxide emission was used to represent the scale effect (*scale*). Additionally, due to the urgent need for quality economic development, the technological effect of FDI should reflect both the production efficiency and environmental optimization and a green total factor productivity (GTPF) can precisely meet these requirements. We therefore used GTPF to represent the technological effect (*tech*) and calculated it using the Malmquist–Luenberger (M-L) indicator of the slack-based measure (SBM) model. Capital, labor and energy consumption are input factors, and output factors include GDP, industrial dust, wastewater, sulfur dioxide and PM2.5 emissions. Among them, the fixed asset stock was used to represent capital and was calculated by the perpetual inventory method [52], which was shown in Equation (2). In Equation (2), $K_t$ and $K_{t-1}$ are the capital stock of phase $t$ and phase $t-1$. $K_0$ is the initial capital stock. $I_t$ is the investment amount of fixed assets in phase $t$. $g$ and $\delta$ are the average annual growth rate of constant investment and fixed assets' depreciation rate, which are calculated by province. $g$ is the geometric average of the growth rate from 2012 to 2018, and the depreciation rate is the arithmetic average. Finally, the proportion of the output value of the secondary industry in gross domestic product (GDP) was used to represent the structural effect (*stru*). The secondary sector plays an irreplaceable role in the economy, but if it is overdeveloped in the pursuit of economic efficiency, it may cause serious environmental problems and thus lead to sustainable urban degradation (Mahmood and others 2020) [53].

$$\begin{cases} K_t = I_t + (1-\delta)K_{t-1} \\ K_0 = I_0\left(\frac{1-\delta}{1+g}\right) \end{cases} \tag{2}$$

### 3.2.3. Control Variables

Control variables include economic growth (*econ*), population size (*pop*), government expenditure (*gov*), society security (*soc*), healthcare (*health*) and education expenditure (*edu*). GDP was used to represent economic growth. The number of people at the end of the year was used to represent the population size. The government's general budget expenditure was used to represent government expenditure. The number of employees in public management, social security and social organizations was used to represent social security. The number of hospitals was used to represent healthcare. The total expenditure of education was used to represent educational expenditure. The data of the variables used in the measurement process and the measurement description of each variable are shown in Table 1.

**Table 1.** Variable meanings and descriptions.

| Variable | Symbols | Measurements |
|---|---|---|
| USC | *usc* | Urban sustainable competitiveness index |
| FDI | *fdi* | The amount of foreign capital actually utilized |
| Scale effect | *scale* | Industrial sulfur dioxide emission |
| technological effect | *tech* | Green total factor productivity |
| Structural effect | *stru* | The output value of the secondary industry in GDP |
| Economic growth | *econ* | GDP |
| Population size | *pop* | The number of people at the end of the year |
| Government expenditure | *gov* | Government's general budget expenditure |
| Social security | *soc* | The number of employees in public management, social security, and social organizations |
| Health care | *health* | The number of hospitals |
| Education expenditure | *edu* | The total expenditure of education |

### 3.3. Sample Data

This analysis takes 282 cities in China from 2012 to 2018 as the research object, with data mainly from the China City Statistical Yearbook. The data for the urban sustainability competitiveness index were sourced from the China Urban Competitiveness Report. Missing values were not processed due to the small number of missing values in the data. Table 2 shows the descriptive statistics for each variable after processing.

**Table 2.** Descriptive statistics.

| Variable | N | Mean | Standard Deviation | Minimum | Maximum |
|---|---|---|---|---|---|
| *usc* | 1972 | 0.326 | 0.148 | 0.001 | 0.989 |
| *lnfdi* | 1861 | 10.14 | 1.862 | 1.099 | 14.94 |
| *lnscale* | 1871 | 10.08 | 1.177 | 0.693 | 13.14 |
| *lntech* | 1974 | 0.243 | 0.675 | 0.059 | 1.081 |
| *lnstru* | 1968 | 3.827 | 0.248 | 2.608 | 4.477 |
| *lnecon* | 1969 | 16.46 | 0.978 | 12.76 | 19.60 |
| *lnpop* | 1972 | 5.882 | 0.699 | 2.986 | 8.133 |
| *lngov* | 1969 | 14.74 | 0.807 | 10.10 | 18.24 |
| *lnsoc* | 1970 | 10.66 | 0.624 | 6.215 | 13.09 |
| *lnhealth* | 1965 | 4.942 | 0.811 | 1.609 | 8.024 |
| *lnedu* | 1867 | 9.968 | 2.171 | 2.996 | 16.14 |

This analysis selects 282 cities in China as the research objective during the period 2012–2018. The empirical study includes three main steps. The first step is to test the impact of FDI on USC. The second step is to test whether FDI has an impact on the USC through a scale effect, technological effect and structural effect, that is, whether the three effects play

a mediating role in the effect of the FDI on USC. The third step is to make further analysis, including endogenous analysis and regional heterogeneity.

## 4. Results and Discussion

### 4.1. Baseline Regression Results

This section begins with an empirical analysis using the least squares pseudo-variance (LSDV) estimation, which has the advantage of being able to obtain the estimates of individual heterogeneity and temporal heterogeneity based on panel data, and to control for measurement error caused by time and individual differences. The specific results are shown in Table 3: column (1) shows the initial impact of FDI on USC, without considering intermediate variables. Only if the results in column (1) are significant is it necessary to further analyze whether there is a mediating effect. Columns (2)–(3) examine the mediating role of the size effect in the impact of FDI on USC, columns (4)–(5) examine the mediating role of the technological effect in the impact of foreign FDI on USC, and columns (6)–(7) examine the mediating role of the structure effect in the impact of FDI on USC.

**Table 3.** Baseline regression results.

| | W=lnscale | | | W=lntech | | W=lnstru | |
|---|---|---|---|---|---|---|---|
| | **(1)** | **(2)** | **(3)** | **(4)** | **(5)** | **(6)** | **(7)** |
| | **Step 1** | **Step 2** | **Step 3** | **Step 2** | **Step 3** | **Step 2** | **Step 3** |
| | *usc* | *lnscale* | *usc* | *lntech* | *usc* | *lnstru* | *usc* |
| *lnfdi* | 0.002 ** | −0.049 *** | 0.001 | −0.008 | 0.002 ** | 0.011 *** | 0.003 ** |
| | (2.020) | (−3.304) | (0.957) | (−0.886) | (1.990) | (3.683) | (2.226) |
| *lnscale* | | | −0.004 ** | | | | |
| | | | (−2.460) | | | | |
| *lntech* | | | | | −0.004 * | | |
| | | | | | (−1.763) | | |
| *lnstru* | | | | | | | −0.022 * |
| | | | | | | | (−1.853) |
| *lnecon* | −0.005 | 0.005 | −0.003 | 0.028 | −0.004 | 0.124 *** | −0.002 |
| | (−0.966) | (0.067) | (−0.609) | (0.952) | (−0.946) | (6.007) | (−0.371) |
| *lnpop* | −0.098 *** | −0.322 | −0.093 *** | 0.180 | −0.097 *** | −0.052 | −0.099 *** |
| | (−5.865) | (−1.201) | (−5.721) | (1.037) | (−5.786) | (−1.094) | (−5.942) |
| *lngov* | −0.007 ** | −0.030 | −0.008 ** | 0.011 | −0.007 ** | 0.032 *** | −0.006 ** |
| | (−2.248) | (−0.541) | (−2.335) | (0.459) | (−2.233) | (3.364) | (−2.023) |
| *lnsoc* | 0.020 *** | 0.159 | 0.027 *** | −0.012 | 0.020 *** | −0.025 | 0.019 ** |
| | (2.615) | (0.681) | (3.514) | (−0.132) | (2.627) | (−0.971) | (2.508) |
| *lnhealth* | 0.000 | 0.019 | −0.000 | 0.017 | 0.001 | −0.016 * | 0.000 |
| | (0.151) | (0.508) | (−0.011) | (0.659) | (0.170) | (−1.943) | (0.041) |
| *lnedu* | −0.001 * | 0.017 * | −0.001 * | 0.008 | −0.001 * | 0.002 | −0.001 * |
| | (−1.847) | (1.756) | (−1.769) | (1.247) | (−1.801) | (1.105) | (−1.778) |
| Constant | 1.563 *** | 11.256 *** | 1.494 *** | −1.556 | 1.557 *** | 0.801 | 1.580 *** |
| | (9.214) | (3.112) | (8.620) | (−0.988) | (9.180) | (1.535) | (9.323) |
| N | 1749 | 1669 | 1668 | 1751 | 1749 | 1750 | 1748 |
| $R^2$ | 0.957 | 0.849 | 0.960 | 0.781 | 0.957 | 0.906 | 0.957 |
| Time effect | Yes | Yes | Yes | Yes | Yes | Yes | Yes |
| Individual effect | Yes | Yes | Yes | Yes | Yes | Yes | Yes |

Note: (i) values in () are *t*-statistics; (ii) ***, ** and * represent 1%, 5% and 10% significance levels, respectively.

First, as shown in Table 3, the coefficient of *lnfdi* in the column (1) is significantly positive, indicating that the higher the inflow of foreign capital, the higher the USC. The co-efficient of *lnfdi* in column (2) is significantly negative, that is, FDI has a negative scale effect, and a large inflow of FDI can reduce urban pollutant emissions, which validates the pollution halo hypothesis [54]. In column (3), the coefficient of *lnfdi* is positive but not significant, while the coefficient of *lnscale* is significantly negative. The significant negative coefficient of *lnscale* shows that environmental pollution is not conducive to the USC, and FDI can affect the USC through the scale effect. The greater the inflow of FDI, the weaker the negative scale effect, that is, the lower the urban pollutant emissions, and the more conducive to improving USC. Moreover, the insignificant coefficient of *lnfdi* shows that the intermediary effect of the scale effect is a fully intermediary effect. Second, the coefficient of *lnfdi* in column (4) is not significant, but the coefficient of *lnfdi* in column (5) is significantly positive, and the coefficient of the *lntech* is significantly negative. This result only suggests that both technological effects and FDI affect USC, but whether technological effects play a mediating role between FDI and USC needs to be further combined with the results of the Sobel's test. The results of the Sobel's test show that the *p*-value is equal to 0.005 and that the mediating effect accounts for about 2.9%, thus confirming a partial mediating role of technological effects. Third, the coefficient of *lnfdi* in column (6) is significantly positive, and the positive structural effect of FDI is obvious. Combined with the significant positive coefficient of *lnfdi* and the significant negative coefficient of *lnstru* in the results of column (7), it can be judged that FDI can partly affect the USC through structural effect. Specifically, as the number of FDI increases, the structural effect becomes more pronounced, but the strong structural effect is not conducive to an increase in USC, mainly because we choose the ratio of secondary industry output to GDP to represent the structural effect. Currently, the FDI in China mainly goes to secondary industries such as manufacturing, electricity and heat production and supply. The higher the volume of FDI, the higher the output value of these industries, and the higher the emissions of carbon dioxide and sulfur dioxide, thus weakening the USC.

Among the control variables, population, government expenditure, social security and education expenditure have a significant effect on USC. The population size has the most significant impact on USC. In addition, social security has a positive impact on USC, while government expenditure and education expenditure have a negative impact on USC, possibly because the expansion of government expenditure and education expenditure can increase the tax burden, hinder capital accumulation and be detrimental to sustainable economic development [55]. Furthermore, although the results indicate that the coefficients of *lngov* and *lnedu* are clearly negative, they are very small and such results do not imply that less government and education spending can be more beneficial to the enhancement of USC, but rather that their quantity should be reasonably controlled.

### 4.2. Endogenous Discussion

In view of the possible endogenous problems in the model, this analysis further uses the two-stage least square method (2SLS to test the impact of FDI on USC. In general, there may be a causal relationship between the explanatory and explained variables. Specifically, FDI affects USC and USC in turn affects FDI. The presence of endogeneity problem increases the statistical error and reduces the precision of the model. This section is therefore analyzed using the 2SLS method, which is a regression analysis method commonly used in economics to address the endogeneity problem in the estimation of causal effects. In the presence of endogeneity, the use of ordinary least squares (OLS) to estimate the model parameters may lead to statistical bias. The 2SLS method uses exogenous variables as instrumental variables to eliminate endogeneity and improve the accuracy of the model through a two-stage regression. In the construction of the 2SLS model, we chosen the lagged period of *lnfdi* as instrumental variables, and the results have been shown in Table 4.

**Table 4.** Endogenous discussion.

| | W=*lnscale* | | | W=*lntech* | | W=*lntech* | |
|---|---|---|---|---|---|---|---|
| | (1) | (2) | (3) | (4) | (5) | (6) | (7) |
| | Step 1 | Step 2 | Step 3 | Step 2 | Step 3 | Step 2 | Step 3 |
| | *usc* | *lnscale* | *usc* | *lntech* | *usc* | *lnstru* | *usc* |
| *lnfdi* | 0.007 *** (3.380) | −0.077 *** (−2.652) | 0.006 *** (2.724) | −0.021 (−1.251) | 0.007 *** (3.355) | 0.006 (0.842) | 0.007 *** (3.356) |
| *lnscale* | | | −0.004 ** (−2.319) | | | | |
| *lntech* | | | | | −0.004 * (−1.805) | | |
| *lnstru* | | | | | | | −0.027 ** (−2.456) |
| Controls | Yes | Yes | Yes | Yes | Yes | Yes | Yes |
| Constant | 1.551 *** (9.824) | 11.230 *** (3.414) | 1.492 *** (9.387) | −1.530 (−1.058) | 1.545 *** (9.798) | 0.812 * (1.694) | 1.573 *** (9.988) |
| N | 1463 | 1388 | 1387 | 1465 | 1463 | 1464 | 1462 |
| $R^2$ | 0.957 | 0.848 | 0.959 | 0.781 | 0.957 | 0.905 | 0.957 |
| Kleibergen–Paap | 166.521 [0.000] | 152.413 [0.000] | 150.718 [0.000] | 167.252 [0.000] | 166.882 [0.000] | 167.226 [0.000] | 168.007 [0.000] |
| Cragg–Donald | 429.661 | 428.373 | 422.748 | 429.611 | 428.837 | 429.252 | 429.082 |
| Time effect | Yes | Yes | Yes | Yes | Yes | Yes | Yes |
| Individual effect | Yes | Yes | Yes | Yes | Yes | Yes | Yes |

Note: (i) values in () are *t*-statistics, and values in [] are *p*-values of the corresponding test statistic; (ii) ***, ** and * represent 1%, 5% and 10% significance levels, respectively. (iii) Kleibergen–Paap test has the original hypothesis that instrumental variables are under-identified, and if the original assumption is rejected then the instrumental variables are identified; Cragg–Donald Wald's F test has the original hypothesis that the instrumental variables are weakly identified. It is reasonable to reject the original hypothesis.

In terms of core independent variables, the coefficient of *lnfdi* in column (1) is significantly positive, indicating that FDI positively enhances the USC, which is consistent with the results in Table 3. The regression results in column (2)–(7) show that FDI has a negative scale effect, but the negative technological effect and positive structural effect of FDI are not significant, and all these effects have a negative impact on USC, again verifying that FDI can affect USC through the scale effect. The Sobel's test in the previous section has verified the mediating role of the technological effect. The Sobel's test for structural effects shows that the *p*-value is equal to 0.01 and the intermediate effect accounts for 3.0% of the total effect, indicating that FDI can influence USC through both technology and structural effects. However, the coefficients of *lnscale* in Table 4 are slightly different from those in Table 3. The coefficient of *lnscale* in Table 3 is positive but not significant, while the coefficient of *lnscale* in Table 4 is positive and significant. This difference suggests that the scale effect based on the endogenous test still has a mediating effect, but the intermediary effect has changed from a full to a partial mediating effect. In terms of control variables, the sign and significance of the coefficients do not change substantially from those in Table 3.

In addition, to test the validity of the instrumental variables, a non-identifiable test (Kleibergen–Paap test) and a weak instrumental variable test (Crag–Donald test) were conducted. In Table 4, the LM value for the Kleibergen–Paap test is significant and the F value for the Crag–Donald test is greater than 10, indicating that the instrumental variables are identifiable and valid. The results of the endogeneity tests above indicate that the sign and significance of the coefficients on the core explanatory and control variables are not

significantly different from those above on the basis of the reasonable validity of the instrumental variables, indicating that there are no endogeneity issues.

### 4.3. Robustness Check

To further test the robustness of the empirical results, this section replaces the explanatory variables and uses the ratio of real utilized foreign capital to GDP to represent FDI in order to re-examine the relationship between FDI and USC. The results are shown in Table 5. In terms of the core explanatory and mediating variables, there is no significant change in the coefficients of each variable. FDI can positively affect USC and the role of scale effect shifts from full to partial mediation, which is consistent with the results in Table 4. In addition, the significance of each variable coefficient is consistent with that in Table 4, although it differs slightly from that in Table 3. In terms of the control variables, the sign and significance of their correlation coefficients slightly change, but the changes are very weak and do not affect the scientific empirical results. In conclusion, the endogenous discussion and robustness tests confirm the reliability of the findings. FDI not only positively affects USC, but also indirectly affects USC through scale, technological and structural effects, thus validating Hypothesis 1 and Hypothesis 2.

**Table 5.** Robustness check.

| | W=*lnscale* | | | W=*lntech* | | W=*lnstru* | |
|---|---|---|---|---|---|---|---|
| | **(1)** | **(2)** | **(3)** | **(4)** | **(5)** | **(6)** | **(7)** |
| | Step 1 | Step 2 | Step 3 | Step 2 | Step 3 | Step 2 | Step 3 |
| | *usc* | *lnscale* | *usc* | *lntech* | *usc* | *lnstru* | *usc* |
| *lnfdi* | 0.001 ** | −0.018 ** | 0.001 * | −0.005 | 0.001 ** | 0.001 | 0.001 ** |
| | (2.119) | (−2.238) | (1.692) | (−1.066) | (2.093) | (0.315) | (2.074) |
| *lnscale* | | | −0.003 ** | | | | |
| | | | (−2.073) | | | | |
| *lntech* | | | | | −0.004 * | | |
| | | | | | (−1.798) | | |
| *lnstru* | | | | | | | −0.025 ** |
| | | | | | | | (−2.132) |
| Controls | Yes | Yes | Yes | Yes | Yes | Yes | Yes |
| Constant | 1.600 *** | 12.680 *** | 1.527 *** | −0.259 | 1.599 *** | 0.881 | 1.622 *** |
| | (9.629) | (4.380) | (8.562) | (−0.177) | (9.596) | (1.429) | (9.741) |
| N | 1749 | 1669 | 1668 | 1751 | 1749 | 1750 | 1748 |
| $R^2$ | 0.957 | 0.848 | 0.959 | 0.781 | 0.957 | 0.905 | 0.957 |
| Time effect | Yes | Yes | Yes | Yes | Yes | Yes | Yes |
| Individual effect | Yes | Yes | Yes | Yes | Yes | Yes | Yes |

Note: (i) values in () are *t*-statistics; (ii) ***, ** and * represent 1%, 5% and 10% significance levels, respectively.

### 4.4. Heterogeneity Discussion

This analysis divides the 282 sample data into the eastern, northeast, central and western regions for regional heterogeneity analysis. Table 6 gives the results of the tests for the mediating effects for each region. First, there are significant regional differences in the impact of FDI on USC. FDI only has a significant positive effect on USC in the eastern region, but in other regions, it does not affect USC nor is there a mediating effect. However, despite this, FDI in the northeast region still has a structural effect, except that this structural effect does not indirectly affect USC, and FDI in the western region has both a scale effect and a structural effect. Moreover, the technological effect and structural effect in the central region can also directly affect USC, while the scale effect in the western region can directly affect USC. Secondly, the results of the tests focusing on the mediating effects in

the eastern region are examined. As the estimated coefficients of *lnscale* and *lntech* are not significant, the analysis of the mediating effects of the scale and technological effects needs to be further combined with Sobel's test, but the structural effects do not need to be tested again with Sobel's test: the coefficient of *lnfdi* in column (6) is significantly negative, while the coefficient of *lnfdi* in column (7) is significantly positive and the coefficient of *lnstru* is significantly negative. According to the results of Sobel's test, the *p*-value is equal to 0.27 when the mediating variable is *lnscale* and 0.74 when the mediating variable is *lntech*. This indicates that FDI in the east cannot affect the USC through the scale and technological effects, but can have an effect through the structural effect.

**Table 6.** Intermediary effects test based on regional differences.

| | Eastern Sample | | | | | | |
| --- | --- | --- | --- | --- | --- | --- | --- |
| | W=lnscale | | | W=lntech | | W=lnstru | |
| | (1) | (2) | (3) | (4) | (5) | (6) | (7) |
| | Step 1 | Step 2 | Step 3 | Step 2 | Step 3 | Step 2 | Step 3 |
| | usc | lnscale | usc | lntech | usc | lnstru | usc |
| lnfdi | 0.014 *** (3.101) | −0.161 *** (−2.639) | 0.011 ** (2.389) | 0.024 (0.879) | 0.014 *** (3.100) | −0.021 *** (−2.967) | 0.012 *** (2.770) |
| lnscale | | | −0.002 (−0.575) | | | | |
| lntech | | | | | −0.001 (−0.244) | | |
| lnstru | | | | | | | −0.081 ** (−2.095) |
| Controls | | | | | | | |
| Constant | 1.064 (1.090) | 12.234 (0.782) | 0.585 (0.597) | −2.925 (−0.506) | 1.061 (1.087) | 2.344 ** (2.239) | 1.225 (1.254) |
| N | 572 | 548 | 548 | 573 | 573 | 572 | 572 |
| $R^2$ | 0.955 | 0.835 | 0.958 | 0.796 | 0.955 | 0.958 | 0.956 |
| Time effect | Yes | Yes | Yes | Yes | Yes | Yes | Yes |
| Individual effect | Yes | Yes | Yes | Yes | Yes | Yes | Yes |
| | Northeast Sample | | | | | | |
| | $W = lnscale$ | | | $W = lntech$ | | $W = lnstru$ | |
| | (1) Step 1 usc | (2) Step 2 lnscale | (3) Step 3 usc | (4) Step 2 lntech | (5) Step 3 usc | (6) Step 2 lnstru | (7) Step 3 usc |
| lnfdi | 0.001 (0.588) | 0.046 (1.500) | 0.000 (0.169) | −0.029 (−1.566) | 0.001 (0.475) | 0.019 *** (3.019) | 0.002 (0.731) |
| lnscale | | | 0.003 (0.751) | | | | |
| lntech | | | | | −0.008 (−1.608) | | |
| lnstru | | | | | | | −0.015 (−0.558) |
| Controls | Yes | Yes | Yes | Yes | Yes | Yes | Yes |
| Constant | 2.156 *** (3.339) | 29.185 ** (2.201) | 1.680 ** (2.588) | −1.294 (−0.216) | 2.146 *** (3.316) | 4.313 * (1.955) | 2.222 *** (3.472) |
| N | 223 | 211 | 211 | 223 | 223 | 223 | 223 |
| $R^2$ | 0.941 | 0.796 | 0.944 | 0.779 | 0.941 | 0.924 | 0.941 |
| Time effect | Yes | Yes | Yes | Yes | Yes | Yes | Yes |
| Individual effect | Yes | Yes | Yes | Yes | Yes | Yes | Yes |

**Table 6.** *Cont.*

| | Central Sample | | | | | | |
|---|---|---|---|---|---|---|---|
| | *W = lnscale* | | | *W = lntech* | | *W = lnstru* | |
| | (1) Step 1 *usc* | (2) Step 2 *lnscale* | (3) Step 3 *usc* | (4) Step 2 *lntech* | (5) Step 3 *usc* | (6) Step 2 *lnstru* | (7) Step 3 *usc* |
| *lnfdi* | 0.004 | 0.030 | 0.005 −0.001 | −0.044 | 0.003 | 0.003 | 0.003 |
| *lnscale* | | | (−0.255) | | | | |
| *lntech* | | | | | −0.008 ** (−2.351) | | |
| *lnstru* | | | | | | | 0.033 * (1.674) |
| Controls | Yes | Yes | Yes | Yes | Yes | Yes | Yes |
| Constant | 0.945 *** (3.964) | 7.738 ** (2.588) | 0.921 *** (3.679) | −5.416 * (−1.919) | 0.901 *** (3.735) | 2.589 *** (2.803) | 0.859 *** (3.423) |
| N | 504 | 482 | 482 | 504 | 504 | 504 | 504 |
| R² | 0.943 | 0.897 | 0.945 | 0.760 | 0.943 | 0.859 | 0.943 |
| Time effect | Yes | Yes | Yes | Yes | Yes | Yes | Yes |
| Individual effect | Yes | Yes | Yes | Yes | Yes | Yes | Yes |
| | Western Sample | | | | | | |
| | *W = lnscale* | | | *W = lntech* | | *W = lnstru* | |
| | (1) Step 1 *usc* | (2) Step 2 *lnscale* | (3) Step 3 *usc* | (4) Step 2 *lntech* | (5) Step 3 *usc* | (6) Step 2 *lnstru* | (7) Step 3 *usc* |
| *lnfdi* | 0.001 (0.612) | −0.056 *** (−3.256) | −0.001 (−0.498) | −0.000 (−0.011) | 0.001 (0.612) | 0.008 * (1.739) | 0.001 (0.690) |
| *lnscale* | | | −0.015 *** (−2.620) | | | | |
| *lntech* | | | | | 0.001 (0.304) | | |
| *lnstru* | | | | | | | −0.018 (−0.804) |
| Controls | Yes | Yes | Yes | Yes | Yes | Yes | Yes |
| Constant | 0.895 *** (3.388) | 12.660 *** (3.203) | 1.076 *** (3.549) | −0.604 (−0.242) | 0.896 *** (3.390) | 1.052 (1.109) | 0.916 *** (3.463) |
| N | 450 | 428 | 427 | 451 | 450 | 451 | 450 |
| R² | 0.954 | 0.938 | 0.956 | 0.808 | 0.954 | 0.888 | 0.954 |
| Time effect | Yes | Yes | Yes | Yes | Yes | Yes | Yes |
| Individual effect | Yes | Yes | Yes | Yes | Yes | Yes | Yes |

Note: (i) values in () are *t*-statistics; (ii) ***, ** and * represent 1%, 5% and 10% significance levels, respectively.

## 5. Conclusions and Policy Suggestions

### 5.1. Conclusions

Using a mediating effects model, this paper investigates the impact of FDI on USC in China in terms of scale, technological and structural effects, using data from Chinese cities during the period 2012–2018. First, FDI has a positive direct effect on USC, similar to the findings that FDI can promote sustainable development [9] and that FDI can improve industrial competitiveness [41]. Aust et al. used a probit model to validate the positive role of FDI in contributing to the achievement of SDGs in the African region [9]. Alvarez and Marin used a systematic Gaussian mixture model (GMM) model to validate the role of FDI in improving industrial competitiveness [41]. However, while these studies identify the role of FDI in sustainable development and industrial competitiveness, there is little analysis of how FDI plays its role. For example, Alvarez and Marin only highlighted the

role of technological innovation in FDI in improving industrial competitiveness, while Aust et al. did not analyze the mechanism of impact and only briefly analyzed the direct role of FDI on sustainable development. Second, FDI can affect USC through scale, technological and structural effects, with the mediating role of the scale effect being the most pronounced. This finding is similar to that of Long et al., who argued that the concept of carbon productivity can reflect the impact of FDI on green development in China. FDI affects carbon productivity through the scale effect, structural effect, technological effect and environmental regulation effect [44]. Third, FDI only positively affects USC in the eastern region and has no significant impact on the northeast, central and western regions. FDI in the eastern region affects USC through the structural effect, while the scale and technological effects do not play a mediating role but both effects can directly affect USC. FDI in the northeast region still has a structural effect, but this structural effect does not indirectly affect USC, while FDI in the western region has both scale and structural effects. In addition, the technology and structural effects in the central region have a direct impact on USC, while the scale effect in the western region has a direct impact on USC. Overall, our study fully affirms the key role of FDI in enhancing USC and the underlying reasons for this effect. Especially in the context of the SDGs, which call for worldwide action, our findings undoubtedly provide new directions to exploit the positive spillover effects of FDI and promote China's sustainable development.

### 5.2. Policy Suggestions

According to the findings of the above study, the following policy implications emerge from this analysis. First, close attention is paid to the three elements of the environment, technology and industry to grasp the new direction of optimizing the quality of FDI. The introduction of FDI should follow the principles of adequacy, effectiveness and rationality. In order to avoid possible negative problems in the process of FDI inflow, it is necessary to achieve a coordinated development of optimized environmental quality, improved technological conditions and balanced industrial structure. In terms of environmental quality, cities should actively expand investment in environmental construction, enforce stricter environmental regulations on local highly polluting enterprises and vigorously promote the concept of energy saving and emission reduction through mandatory management and spontaneous publicity. At the same time, cities should strengthen the cultivation of clean technologies, develop more new sources of energy as soon as possible, and make scientific and efficient use of existing clean technologies when large amounts of FDI enter, thus reducing pollutant emissions. In terms of technological innovation, cities should actively expand their financial expenditure on science and technology and adhere to the path of independent research and development. In addition, cities should pay attention to developing the skills and improving the quality of workers in local enterprises, and increase support for these enterprises to avoid exclusion and brain drain when FDI enters. In terms of industrial structure, cities should adjust the industrial structure of FDI and increase the scale of FDI in tertiary industries such as information transmission, software and information technology services, scientific research and technical services.

Secondly, the introduction of FDI should follow the principles of differentiation and diversification, and take into account regional characteristics. The northeast and central and western regions have not yet developed a significant FDI promotion effect on USC. Therefore, these regions should continue to focus on increasing the amount of FDI. However, FDI in the eastern region is entering a critical period of transition towards quality optimization. The eastern region should increase the rate of optimizing the quality of FDI by significantly increasing investment in research and education, while maintaining the current total amount of FDI. In addition, the central region should continue to strengthen the supporting role of the secondary industry and bring into play the driving role of the secondary industry in improving the USC. In the western region, optimizing the quality of the environment and reducing the emissions of pollutants such as sulphur dioxide

is a primary consideration, which is particularly important for improving USC in the western region.

*5.3. Further Study*

While our study has made some interesting findings regarding the relationship between FDI and USC, there are still some limitations that need further investigation. For example, this analysis directly used the sustainable competitiveness index to measure USC without disaggregating the discussion of the dimensions of the index. Future research could focus on the analysis of USC heterogeneity to derive a comprehensive picture of the relationship between FDI and USC across countries and regions. In addition, our research focused on how FDI affects USC through technological effects, but the technological effects were only analyzed in terms of production efficiency. Future research should explore more effects, such as competitive effects, demonstration effects, industry linkage effects, etc.

**Author Contributions:** J.L.: conceptualization, methodology and software. L.J.: data collection and writing. B.J.: writing, reviewing and editing. S.L.: reviewing and editing. All authors have read and agreed to the published version of the manuscript.

**Funding:** This research was funded by [Natural Science Foundation of Shandong Province] grant number [ZR2020MG044] and [ZR2022MG042].

**Institutional Review Board Statement:** Not applicable.

**Informed Consent Statement:** Not applicable.

**Data Availability Statement:** The datasets used and/or analyzed during the current study are available upon reasonable request.

**Conflicts of Interest:** This article has no interest or conflict with any individual or group. The authors declare that they have no competing interests.

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
