# Peer review of "How Does FDI Enhance Urban Sustainable Competitiveness in China?"

_sustainability, doi:10.3390/su151310393_

Round 1

Reviewer 1 Report

1)      Abstract should be revised and especially some main findings of the study should be added.

2)      Please remark on specific objectives clearly.

3)      Introduction section should be revised. the literature review should be enlarged and some key results from the literature should be presented. The introduction should be prepared more significantly.

4)      System modeling section should be revised.

5)      Results need to be compared with experimental works from the literature especially here results from simulation.

6)      Authors should revise all of the studies and be careful in giving all explanations with clear and easy words for better understanding

7)      Some key findings should be added to the conclusion section.

8)     In the conclusions section, I would like to say that a result sentence is required for the possible use of this paper.

-----

Reviewer 2 Report

Overall, I found this paper interesting and illuminating. That being said, I have two minor recommendations prior to acceptance for publication.

The first recommendation is to add some detail to better differentiate the overall analysis from that of the intermediaries in the tables (Tables 3, 4, 5, and 6).  Having four columns in the results table with the same heading -- USC (Urban Sustainable Competitiveness) – is confusing.  And while the discussion of the Baseline results (Section 5.1) attempts to explain how the pairs of variables function together – and have different meanings – after the initial general indicator in the first column, perhaps using subscripts or another means of highlighting the different meanings of the indicator would minimize (if not eliminate) confusion on the part of those trying to understand the results.

The other recommendation is to “spell out” the term SDG – Sustainable Development Goals – rather than use the acronym on Line 126 – the only time the term appears in the paper.  The acronym would remain as part of the reference article title but is not needed in the paper’s text.

Reviewer 3 Report

The study is a good attempt however the authors should carefully address the following suggestions:

1. The study should add a starting paragraph on the importance of FDI on sustainable unban competitiveness.

2. The novelty of the study is not sufficiently clarified, the paper did highlight the contribution of the paper in the literature, but it is quite weak. It is important to identify the weakness of previous papers and show the argument and contribution of the paper.

3. It would be great if this study adds some more latest statistics on the Chinese economy concerning FDI and urban sustainable competitiveness.

4. The study should provide justification for using the methods used in the study, and how it is better than others from the literature.

5. The study should describe the seminal work about the association between FDI and sustainable urban competitiveness.

6. The study literature review is not very strong and needs to update till 2023, and cite the mixed studies evidence in the literature summary paragraph.

7. The study needs to describe the theoretical relationship between FDI and sustainable urban competitiveness based on some theory and build hypotheses.

8. Conclusion and policy recommendations and future study directions should be written as separate headings.

9. English grammar should be checked properly.

English grammar should be checked properly.

Round 2

Reviewer 1 Report

——-

——

Reviewer 3 Report

The authors have significantly revised the manuscript.